# Rat Hepatic Stellate Cell Line CFSC-2G: Genetic Markers and Short Tandem Repeat Profile Useful for Cell Line Authentication

**DOI:** 10.3390/cells11182900

**Published:** 2022-09-16

**Authors:** Indrajit Nanda, Sarah K. Schröder, Claus Steinlein, Thomas Haaf, Eva M. Buhl, Domink G. Grimm, Ralf Weiskirchen

**Affiliations:** 1Institute of Human Genetics, Julius Maximilians University of Würzburg, D-97074 Würzburg, Germany; 2Institute of Molecular Pathobiochemistry, Experimental Gene Therapy and Clinical Chemistry (IFMPEGKC), RWTH University Hospital Aachen, D-52074 Aachen, Germany; 3Electron Microscopy Facility, Institute of Pathology, RWTH Aachen University Hospital, D-52074 Aachen, Germany; 4Campus Straubing for Biotechnology and Sustainability & Weihenstephan-Triesdorf University of Applied Sciences, Technical University of Munich, D-94315 Straubing, Germany

**Keywords:** liver, extracellular matrix, hepatic stellate cell, myofibroblast, fibrosis, stress fibers, spectral karyotyping, rhodamine–phalloidin stain, next-generation sequencing, STR profile

## Abstract

Hepatic stellate cells (HSCs) are also known as lipocytes, fat-storing cells, perisinusoidal cells, or Ito cells. These liver-specific mesenchymal cells represent about 5% to 8% of all liver cells, playing a key role in maintaining the microenvironment of the hepatic sinusoid. Upon chronic liver injury or in primary culture, these cells become activated and transdifferentiate into a contractile phenotype, i.e., the myofibroblast, capable of producing and secreting large quantities of extracellular matrix compounds. Based on their central role in the initiation and progression of chronic liver diseases, cultured HSCs are valuable in vitro tools to study molecular and cellular aspects of liver diseases. However, the isolation of these cells requires special equipment, trained personnel, and in some cases needs approval from respective authorities. To overcome these limitations, several immortalized HSC lines were established. One of these cell lines is CFSC, which was originally established from cirrhotic rat livers induced by carbon tetrachloride. First introduced in 1991, this cell line and derivatives thereof (i.e., CFSC-2G, CFSC-3H, CFSC-5H, and CFSC-8B) are now used in many laboratories as an established in vitro HSC model. We here describe molecular features that are suitable for cell authentication. Importantly, chromosome banding and multicolor spectral karyotyping (SKY) analysis demonstrate that the CFSC-2G genome has accumulated extensive chromosome rearrangements and most chromosomes exist in multiple copies producing a pseudo-triploid karyotype. Furthermore, our study documents a defined short tandem repeat (STR) profile including 31 species-specific markers, and a list of genes expressed in CFSC-2G established by bulk mRNA next-generation sequencing (NGS).

## 1. Introduction

During the last years, immortalized HSC lines have become an established experimental tool in biomedical research in place of primary cells. Compared to primary cells, immortalized HSC lines offer several advantages. They are cost effective, easy to handle, and provide an unlimited supply of biological material. Moreover, they can bypass ethical concerns that may arise by isolation of primary cells from animal or human tissue [1]. In addition, HSC lines have a clonal origin and are biologically homogenous, allowing the performance of more reproducible experiments when compared to primary HSCs that constitute a highly heterogeneous cell population [2].

Presently, continuous HSC lines were established from mouse, rat, pig, and humans. Although widely applied, only several of these continuous lines have been genetically characterized, while numerous scientific journals and funding agencies now request cell authentication before considering publication or funding when working with cell lines. Some years ago, we described genetic characteristics of the human HSC line LX-2 [3]. In addition, we recently published genetic information about the two murine cell lines GRX and Col-GFP HSC and the rat HSC line HSC-T6 [4,5,6].

Nevertheless, based on the fact that results obtained from one specific cell type might require the reproduction in another cell line, we here established genetic characteristics for the cirrhotic fat-storing cell line (CFSC). This cell line and four different clonal sublines derived thereof, i.e., CFSC-2G, CFSC-3H, CFSC-5H, and CFSC-8B, were originally established in the laboratory of Marcus Rojkind from a carbon chloride cirrhotic rat liver and proposed as a useful tool to study extracellular matrix production by HSCs [7,8].

In the present study, conventional chromosome analysis and multicolor spectral karyotyping were accomplished to gain insight into the genetic organization of CFSC-2G cells that has successfully uncovered numerous characteristic features of its chromosome organization. We further established a specific short tandem repeat (STR) profile for this cell line that includes 31 species-specific allelic variant sites. Bulk sequencing of mRNA isolated from CFSC-2G cells confirmed the expression of typical HSC markers associated with extracellular matrix synthesis and turnover. In addition, electron microcopy and Rhodamine–Phalloidin stain confirmed that CFSC-2G cells form a robust microfilament network, cytoplasmic fat droplets, and other cellular features typical for HSCs. In sum, our study provides genotypic and phenotypic characteristics of CFSC-2G cells that are useful for cell authentication.

## 2. Materials and Methods

### 2.1. Literature Search

Papers using CFSC cells were identified by searching the PubMed database [9] using the search term “CFSC”, excluding hits that used the abbreviation for other terms such as “cese del flujo sanguíneo cerebral”, “Cancer Fatigue Scale-Chinese”, “chronic severe functional constipation”, “chyle fat-derived stem cells”, “cisplatin-loaded fibers/sponge composite”, “clinical frailty score based on chart review data”, “class-driven feature selection and classification model”, “cell-free supernatant of the co-culture”, “chloroform“, “consideration of future safety consequences”, “calculated free serum cortisol”, “Chicago Food System Collaborative”, “circumferential fiber shortening”, “Community and Family Study Center”, “Current Fetal Surgery Center”, and “Chinese version of CFS”. Additional papers were identified by a standard Google search using the search terms “CFSC” and “Rojkind” or “CFSC” and “liver” or “CFSC” and “Greenwel”. All papers related to the continuous lines CFSC, CFSC-2G, CFSC-3H, CSFC5H, and CFSC-8B identified in our searches are listed in Appendix A.

### 2.2. Cell Culture

CFSC cells were established and characterized over thirty years ago in the laboratory of Marcus Rojkind [7,8]. It is listed in the Cellosaurus database under accession no. CVCL_M104 as a spontaneously immortalized HSC cell line derived from adult male Wistar rats. From this parenteral cell line four clonal cell lines were derived, namely CFSC-2G (CVCL_4U34), CFSC-3H (CVCL_4U35), CFSC-5H (CVCL_4U36), and CFSC-8B (CVCL_4U37). CFSC-2G cells used in this study were routinely propagated in 10 cm Petri dishes and cultured in Dulbecco’s modified Eagle’s medium (DMEM) supplemented with 10% fetal bovine serum (FBS), 2 mM L-Glutamine, 1 mM sodium pyruvate, 1 × Penicillin/Streptomycin, and 1 × non-essential amino acids. Medium exchange was conducted every second day and cells were subcultured using Accutase solution (#A6964, Sigma-Aldrich). WI-38 (VA-13) (#CCL-75.1, ATCC, Manassas, VA, USA) and HSC-T6 cells [10] were cultured in the same medium as CFSC-2G cells excluding non-essential amino acids.

### 2.3. Electron Microscopic Analysis

Electron microscopic analysis of CFSC-2G cells was essentially conducted as described before [6]. In brief, cells were fixed in 1× phosphate buffered saline (PBS) containing 3% glutaraldehyde, washed in 0.1 M Soerensen’s phosphate buffer (Merck, Darmstadt, Germany), and fixed in a solution of 1% osmium tetroxide (OsO_4_) (Roth, Karlsruhe, Germany) solved in 25 mM sucrose buffer (Merck). Fixed cells were dehydrated and subsequently incubated in propylene oxide (Serva, Heidelberg, Germany) in a mixture of Epon resin (Serva) and propylene oxide (1:1) and pure Epon. Finally, ultrathin sections (70–100 nm) were prepared and packed upon Cu/Rh grids (HR23 Maxtaform, Plano GmbH, Wetzlar, Germany). Contrast was enhanced by staining with 0.5% uranyl acetate and 1% lead citrate (both Science Services, Munich, Germany). The samples were analyzed at an acceleration voltage of 60 kV using a Zeiss Leo 906 (Carl Zeiss AG, Oberkochen, Germany) transmission electron microscope and depicted images were taken at magnifications of 2156× to 35,970×.

### 2.4. Preparation of CFSC-2G Metaphase Chromosomes and Karyotyping

The preparation of chromosomes of CFSC-2G was essentially conducted as published before [6]. In brief, semi-confluent CFSC-2G cultures were exposed to colcemid solution (Gibco, ThermoFisher Scientific, Dreieich, Germany), detached, and harvested by centrifugation. Thereafter, the cells were treated with 0.56% (*w*/*v*) hypotonic potassium chloride solution and fixed with a mixture of methanol and acetic acid (3:1). Air-dried chromosome spreads were prepared, and slides were treated with 0.025% (*w*/*v*) trypsin solution followed by staining with Giemsa solution to visualize G-banding pattern. In addition, the heterochromatin in metaphase chromosomes was stained following an established procedure [11]. At least ten metaphases were analyzed from the GTG-stained (G-bands by trypsin using Giemsa) and CBG-stained (C-bands by Barium hydroxide using Giemsa) slides.

### 2.5. In Situ Hybridization and Spectral Imaging

Detection of rearrangements in the CFSC-2G karyotype was performed using a commercially available rat SKY probe (Applied Spectral Imaging Inc., Carlsbad, CA, USA). Denaturation of SKY probes, in situ hybridization, chromosomal counterstaining, embedding, spectral imaging, and analysis of multicolor-stained chromosomes were conducted following a protocol described before [6,12].

### 2.6. Short Tandem Repeat (STR) Profiling

STR profiling and interspecies contamination test for CFSC-2G cells was performed using the cell line authentication service from IDEXX (Kornwestheim, Germany) using the CellCheck^TM^ Rat system that includes 31 species-specific STR markers.

### 2.7. Next-Generation Sequencing and Data Analysis

RNA from CFSC-2G cells was isolated by CsCl density gradient centrifugation as described before [6]. Therefore, cells grown to 80% confluence were lysed and homogenized in a guanidine thiocyanate-containing buffer. The cell lysate was layered onto a cesium chloride cushion and centrifuged in an Optima L-90K ultracentrifuge (Beckman Coulter, Inc., Krefeld, Germany). The resulting RNA pellet was re-suspended in sterile water, once purified by ethanol precipitation, and finally re-suspended in sterile water. The concentration, purity, and quality were determined by UV spectroscopy and on the Agilent 4200 TapeStation automated platform (Agilent Technologies Inc., Waldbronn, Germany). Depletion of ribosomal RNAs, library preparation, sequencing, and bioinformatics analysis were essentially performed as described before using the ENSEMBL data set of the rat genome (release 104) [6]. The abundance of individual gene transcripts obtained by bulk sequencing is given in Transcripts Per Million (TPM).

### 2.8. Western Blot Analysis

Preparation of protein extracts, protein quantification, and Western blot analysis was performed using established protocols [13]. Equal protein amounts (40 µg/lane) from CFSC-2G and WI-38 (VA-13) cells were heated at 80 °C for 10 min and separated in 4–12% Bis-Tris gels (Invitrogen, Darmstadt, Germany) under reducing conditions using MES running buffer. Separated proteins were electro-blotted on nitrocellulose membranes (Schleicher & Schuell, Dassel, Germany) and the transfer controlled by Ponceau S stain. Thereafter, unspecific binding sites were blocked and membranes probed with antibodies specific for Fibronectin, SV40 large T antigen (SV40T), α-smooth muscle actin (α-SMA), collagen type I, and Vimentin. The blots were further probed with antibodies directed against Glyceraldehyde-3-phosphate dehydrogenase (GAPDH) and β-actin to document integrity of separated proteins. Primary antibodies were detected with horseradish-peroxidase (HRP)-conjugated secondary antibodies and the Supersignal™ chemiluminescent substrate (Perbio Science, Bonn, Germany). Sources and specificities of primary and secondary antibodies used are depicted in Table 1.

### 2.9. Rhodamine–Phalloidin Stain

Microfilaments in CFSC-2G cells were stained with a Rhodamine–Phalloidin conjugate following essentially a protocol described before [6]. The stained cells were washed and nuclei counterstained by incubation in a 4′,6-diamidino-2-phenylindole (DAPI) solution (#D1306, ThermoFisher Scientific). Finally, the glass coverslips were mounted with PermaFluor aqueous mounting medium (#TA-030-FM, ThermoFisher Scientific) and analyzed under a Nikon Eclipse E80i fluorescence microscope equipped with the NIS-Elements Vis software (version 3.22.01, Nikon Europe B.V., Amstelveen, The Netherlands).

### 2.10. Lipid Droplet Staining

Stimulation of CFSC-2G with oleic acids and subsequent staining with BODIPY^TM^ 493/503 to visualize lipid droplets was performed essentially as reported before [6]. The stained cells were analyzed using a Nikon Eclipse E80i microscope (Nikon Europe, Düsseldorf, Germany).

## 3. Results

### 3.1. Usage of CFSC in Biomedical Research

The cell line CFSC and its derivatives were originally established about three decades ago in the laboratory of Marcos Rojkind from carbon-tetrachloride-induced cirrhotic rat liver [7,8]. Dr. Rojkind (Figure 1), who established that cell model for biomedical research, was a distinguished expert on molecular mechanisms underlying the pathogenesis of alcohol-induced liver fibrosis and cirrhosis with special interest into the role of hepatic stellate cells in these processes.

Actually, there are approximately 100 peer-reviewed studies available reporting experimental findings obtained in parental CFSC and/or its derivative subclones (i.e., CFSC-2G, CFSC-3H, CFSC-5H, and CFSC-8B) (Appendix A).

After establishment in year 1991, studies are regularly published each year that rely on the usage of CFSC (Figure 2A). Most of these studies are performed in China, the USA, Mexico, and Germany, and 30% of all studies that used CFSC were conducted as joint activities of laboratories located in different countries (Figure 2B, Appendix A). In addition, data on experimental findings with respective cells were presented in many international meetings and are published in a wealth of meeting abstracts and doctoral dissertations.

### 3.2. Phenotypic Characteristics of CFSC-2G Cells

In culture, CFSC-2G cells have a typically fusiform fibroblastic morphology, which is in agreement with the proposed fibrogenic phenotype (Figure 3). When cultured at low density, the cells possess processes extending out from the ends of the cell body, while when cultured at higher density the cells form a well-structured dense cell layer.

### 3.3. Ultrastructural Analysis of CFSC-2G

CFSC-2G cells have ultrastructural features that are typical for HSCs, including large nuclei, striking endolysomal vesicles, prominent Golgi apparatus, remarkable rough endoplasmic reticulum, well-pronounced fat droplets, large mitochondria, and dense glycogen granules, which can be detected by means of transmission electron microcopy (Figure 4).

### 3.4. Expression of Typical Hepatic Stellate Cell Markers in CFSC-2G

The cell line CFSC was originally established from adult male Wistar rat cirrhotic liver and shown to express typical HSC markers including collagens (type I, III, and IV), Fibronectin, and laminin [7,8]. In agreement with this former characterization and the proposed ancestry from HSC, we could confirm the expression of typical HSC markers including Fibronectin, α-smooth muscle actin (α-SMA), collagen type I, and Vimentin. Moreover, the cells were negative for SV40T that was used for immortalization of many other continuous HSC lines, such as LX-1, LX-2, HSC-T6, SV68-IS, A640-IS, IMS/MT (-), IMS/N, and Col-GFP HSC (Figure 5) [10,15,16,17,18,19].

Moreover, typical for HSCs that have undergone myofibroblastic differentiation, cultured CFSC-2G cells form a robust network of microfilaments that can be visualized by Rhodamine–Phalloidin (Figure 6).

### 3.5. Fat Uptake and Storage in CFSC-2G

Another important feature of HSC, earlier known as fat-storing cells, is the capacity to take up and accumulate lipids in lipid droplets. These intracellular, fat-storing organelles serving as neutral lipid reservoirs and the center of lipid metabolism can be easily colored by fluorescent dyes interacting with extremely hydrophobic substances and visualized by fluorescence microscopy including Nile Red or by different BODIPY-based dyes [20,21]. In particular, HSCs cultured in the presence of oleic acid form large intracellular lipid droplets and increase the expression of lipid-droplet proteins [22].

To confirm that moderate administration of oleic acid can increase intracellular lipid droplet diameter, we comparatively analyzed the content of lipid droplets in cultured CFSC-2G cells that were treated with oleic acid or vehicle (i.e., phosphate-buffered saline solution). This analysis revealed that the continuous line CFSC-2G, like HSC-T6, has the propensity to accumulate oleic acid within intracellular droplets, again supporting one of the metabolic hallmarks of HSCs, namely that the supplementation with fatty acids in activated HSCs or transdifferentiated myofibroblasts is associated with increased formation of lipid droplets (Figure 7) [6,23].

We next isolated mRNA from CFSC-2G cells cultured under basal conditions to 80% confluence and performed bulk mRNA sequencing (mRNA-Seq). The identity and abundance of each mRNA were determined using the *Rattus norvegicus* reference transcriptome. The complete list of mRNAs detected with their abundances is given in Appendix A. In total, we found 22,827 different transcript species with abundances ranging from 3.5 × 10^−11^ (ENSRNOT00000045400.5, *Macf1*, microtubule-actin crosslinking factor 11) to 12,768.5 (ENSRNOT00000067875.4, *Spp1*, secreted phosphoprotein 1). In particular, we found strong expression of α-smooth muscle actin (*Acta2*, TPM: 10.9245), collagen type I α1 (*Col1a1*, TPM: 1408.42), collagen type III α1 (*Col3a1*, TPM: 644,932), Vimentin (*Vim*, TPM: 3853.69), two Fibronectin 1 transcripts (*Fn1*, TPM: 1383.81 and 702,835), galectin 1 (*Lgals1*, TPM: 8557.92), secreted protein acidic and cysteine rich (*Sparc*, TPM: 1040.23), tissue inhibitor of metalloproteinases 1 (*Timp1*, TPM 714,162), *Timp2* (TPM: 1030.84), Cd63 (TPM: 2732.07), transgelin 2 (*Tagln2*, TPM: 642,605), cofilin 1 (*Cfl1*, TPM: 1961.22), caveolin 1 (*Cav1*, TPM: 486,234), and cysteine and glycine-rich protein 2 (*Csrp2*, TPM: 62.7395), respectively. All these markers indicate that CFSC-2G cells originate from primary HSCs. The expression of most of these markers was recently demonstrated in rat HSC line HSC-T6 cultured under the same culture conditions (Table 2). Furthermore, CFSC-2G cells express all three retinoid nuclear receptors (RARα, RARβ, and RARγ), the three retinoid X receptors (RXRα, RXRβ, and RXRγ), and two different transcripts of the antiviral response tetherin (*Bst2*), representing a component of the innate immune response to enveloped viruses. Similar to HSC-T6, the NGS data demonstrated that CFSC-2G cells express large quantities of ferritin heavy chain 1 (*Fth1*, TPM: 5047.69), ferritin light chain 1 (Ftl1, TPM: 3658.31), and ferritin light chain 1-like protein (LOC100360087, TPM: 5451.91), which are associated with the intracellular storage of iron. Moreover, in line with their activated phenotype, CFSC-2G cells express only low mRNA amounts of the peroxisome proliferation-activated receptor-γ (*Pparg*), representing one nuclear receptor whose activity is significantly decreased in activated HSCs [24].

### 3.6. Genetic Analysis of CFSC-2G Cells

#### 3.6.1. Conventional Cytogenetic Analysis of CFSC-2G Cells

Enumeration of Giemsa-stained chromosomes of CFSC-2G cells at different intervals consistently showed a shift from the standard diploid chromosome number that could be characterized as a hyperdiploid karyotype with varying number of chromosomes ranging from 53 to 65 (Figure 8). No cell with normal diploid chromosomes was observed. However, around 50% of the total 60 analyzed cells appeared to be near triploid with 62–63 chromosomes. Remarkably, some chromosomes in these triploid cells were unusually large and bi-armed, pointing to structural rearrangements. An attempt to identify specific chromosomes with plausible rearrangements was unsuccessful, as the banding pattern and presence of numerous smaller chromosomes imposed an enormous challenge to classify chromosomes accurately. Nevertheless, a high level of chromosome rearrangements in CFSC-2G cells could be inferred from the GTG-banded metaphases (cf. Figure 8B).

Consistent with the GTG-banded chromosomes, large bi-armed rearranged chromosomes were noted in the CBG-banded metaphase plates (cf. Figure 8A). Though the heterochromatin is confined to the centromeric regions, traces of heterochromatic region were visualized interstitially in some of these large, rearranged chromosomes. Significantly, the CBG banding was able to identify a single complete heterochromatic Y chromosome, indicating that the CFSC-2G cell was originally derived from a male rat. Noteworthy, this chromosome did not undergo rearrangement compared to other chromosomes.

Strikingly, cytogenetic analysis in CFSC-2G cells further detected the sporadic occurrence of ring chromosomes in metaphases that appeared to differ in size and were partly di-centric (Figure 8C–E). In total, 11% of the analyzed cells showed distinct ring chromosomes.

#### 3.6.2. In Situ Hybridization and Spectral Karyotype Analysis

Due to the specific color code assigned to individual chromosomes, the SKY analysis precisely allows authentic identification of both structural and numerical aberrations. Figure 9 displays a metaphase spread and karyotypes exhibiting clear SKY hybridization signals, allowing the specification of individual chromosomes along with information about numerical and structural aberrations. The representative SKY analysis image displays extensive rearrangement of the CFSC-2G karyotype both numerically and structurally. For instance, most chromosomes occur in multiple copies along with several derivative chromosomes resulting from the translocation of non-homologous chromosome segments.

However, the most striking observation was found for chromosome 1 (RNO1), which was found in five copies. All RNO1 copies are literally derivative ones in which the tiny, short arms are replaced by different chromosome segments, hinting to chromosome translocations and simultaneous loss of the short arm (*p*-) material in all these RNO1 derivatives. Both RNO2 and RNO7 are organized in multiple copies, but these two autosomes displayed unusual organization for a bi-armed chromosome that may arise through isochromosome formation, pointing to an overt duplication of the long arms for RNO2 and RNO7. Additionally, the SKY analysis disclosed translocations of different chromosomal segments in the derivative chromosomes of RNO2, RNO4, RNO6, RNO10, RNO14, RNO17, and RNO19. Both RNO9 and RNO18 are the only ones that are organized in two copies. On the other hand, three chromosomes (i.e., RNO11, RNO 12, and RNO 20) are present in two copies, but the SKY analysis revealed that additional parts of these chromosomes were also found in other derivative chromosomes. Likewise, RNO15 and RNO16 occurred in single copy, but multiple segments from these two chromosomes were found in different derivative chromosomes.

A summary of the recurrent occurrence of different translocations revealed through SKY hybridization is shown in Table 3. Furthermore, one particular observation was that beside the single Y chromosome, the SKY analysis identified two X chromosomes.

#### 3.6.3. Short Tandem Repeat Analysis

The karyotype analysis showed that CFSC-2G cells are characterized by extensive numerical and structural rearrangements. To establish simpler characteristics to uniquely identify CFSC-2G cells, we next performed short tandem repeat (STR) DNA profiling using a set of 31 established polymorphic markers. The resulting PCR products were analyzed on a genetic analyzer resulting in a well analyzable electropherogram for each marker tested (Appendix A) that allowed to define a characteristic allele constitution for each of the 31 mouse-specific markers. The results of this analysis showed that CFSC-2G had a unique STR profile that was markedly different from that we recently established for rat HSC line HSC-T6 (Table 4).

## 4. Discussion

Continuous HSC lines resemble primary cultured HSCs in many ways and are valuable and useful tools to address issues of HSC biology and function. As a result, the number of studies which use HSC lines is steadily increasing. However, from the ~30 HSC lines available, only a few lines have found widespread use [1]. In regard to cells originated from rats, two lines have presently attracted the most interest. The most widely used line, HSC-T6, was established by transfection of primary HSC from male Sprague-Dawley rats with a cDNA containing SV40T [10]. We recently characterized this cell line in regard to genetic features that can be used for cell authentication [6]. The second line, termed CFSC, and its subclones CFSC-2G, CFSC-3H, CFSC-5H, and CFSC-8B, were obtained over 30 years ago from spontaneous immortalization of primary HSCs that were isolated from CCl_4_ cirrhotic liver [7,8]. Since then, this cell line has been used to address general issues of HSC biology. In particular, the clone CFSC-2G has obtained widespread use in many laboratories all over the world that have interest to understand the genetic and epigenetic aspects and mechanisms of hepatic fibrosis (cf. Appendix A).

Nevertheless, the chromosomal and genetic characteristics of this cell line were not established. It is well accepted that continuous cell lines are under permanent stress and selection due to the artificial environment in which they exist. Genetic mutation can occur during cell passaging and laboratory manipulation, which can lead to a heterogeneous population with a selective advantage. As a consequence, this could result in a final cell population that is completely different from the starting pool. Unfortunately, the factors provoking a genetic drift within a cell line can be different in various labs.

A fundamental study has shown that established cell lines, generally thought to be clonal, are highly genetically heterogeneous, most likely resulting from clonal dynamics and continuous instability [25]. It is supposed that these factors are the reason why findings involving established cell lines are often difficult to reproduce [25]. In addition, there is growing evidence that cell line misidentification is a fundamental problem in biomedical science that gives reasons for errors, false conclusions, and irreproducible experiments [26].

Consequently, it is necessary to identify resilient biochemical and genetic features of a continuous growing cell line to confirm the identity and genetic stability and to mark the ‘biochemical quality’.

In our study, we describe several molecular features of CFSC-2G cells that are useful for cell authentication. The cell line has a fibroblastic morphology that is in agreement with the proposed fibrogenic phenotype (Figure 3). Electron microscopy showed that CFSC-2G cells have ultrastructural features that are typical for HSCs, including large nuclei, pronounced endolysomal vesicles, prominent Golgi complexes, remarkable rough endoplasmic reticulum, intracellular lipid droplets, large mitochondria, and glycogen granules (Figure 4). Furthermore, Western blot analysis showed that the cells are capable of expressing typical HSC markers such as Fibronectin, Vimentin, and α-SMA, while expectedly lacking expression of SV40T (Figure 5). These biochemical features underpin that the cells are of HSC origin and lack SV40T that was used for immortalization of many other HSC lines (e.g., LX-1, LX-2, HSC-T6, SV68-IS, A640-IS, IMS/MT (-), IMS/N, and Col-GFP HSC) [10,15,16,17,18,19].

Furthermore, CFSC-2G cells have a prominent network of intracellular actin filaments (Figure 6) and are capable of uptaking and accumulating fat in intracellular lipid droplets (Figure 7) that are other important hallmarks of HSCs, physiologically acting as professional retinoid-storing cells [1].

Bulk mRNA sequencing revealed that similar to rat HSC-T6 cells, CFSC-2G cells express large mRNA quantities of *Acta2*, *Col1a1*, *Col3a1*, *Vim*, *Fn1*, *Lgals1*, *Sparc*, *Csrp2*, *Timp1*, *Timp2*, *Tagln2*, retinol-binding proteins, and their nuclear receptors (Table 2). Again, these markers recapitulate many activated HSC markers that are often used as markers that become increasingly expressed during hepatic fibrosis [27]. Bulk mRNA sequencing further revealed that CFSC-2G cells express only low amounts of *Pparg* mRNA, which is in agreement with an activated HSC phenotype [24].

Previous work demonstrated that the treatment with all-trans retinoic acid (ATRA) inhibits proliferation and diminishes mRNA expression of procollagen α1(I), procollagen α1(III), transforming growth factor-β1, connective tissue growth factor (*Ctgf/ccn2*), matrix metalloproteinase-2 (*Mmp2*), *Timp1*, *Timp2*, and plasminogen activator inhibitor-1 (PAI-1/*Serpine1*) in CFSC-2G [28]. Our analysis confirmed that all these profibrogenic marker genes are expressed at a high level in CFSC-2G under normal culture conditions without ATRA supplementation, once again underpinning the activated phenotype of this continuous growing HSC line (cf. Table 2 and Appendix A).

Each of the ~30 HSC cell lines available has characteristic features making them useful for different investigations [1]. However, all lines have advantages and disadvantages, and the choice of which cell line is best suited for a specific study depends on many factors. First, it relies on the species (mouse, rat, or human) in which the experiments should be conducted. Second, some countries have regulatory restrictions when working with genetically modified cells carrying transgenes, such as SV40T (e.g., SV68-IS, A640-IS, IMS/MT, IMS/N, Col-GFP HSC, HSC-T6, and LX-1) or the human telomerase reverse transcriptase (e.g., TWNT-1, hTERT-HSC, and NPC-hTERT) [1]. Restriction might also be considered when working with the murine cell line GRX, which produces a large amount of retroviruses [4]. Third and most important, the repertoire of genes expressed in a chosen cell line must be taken into account. If someone, for example, will analyze aspects of lipid homeostasis or turnover, they should be sure that the chosen line expresses all relevant genes required for respective analysis. However, continuous growing HSC lines still have many differences compared to primary HSCs, particularly when aspects of quiescence should be analyzed, that cannot be properly mimicked in an immortalized cell line [1].

The chromosomal changes detected through SKY analysis distinctly revealed numerical and extensive chromosome rearrangements in CFSC-2G cells. In a few metaphases we observed different sizes of ring chromosomes, including di-centric ones, which can contribute to a general genomic instability (Figure 8). By SKY analysis we further identified 13 different detectable translocations, 3 duplications of complete arms, and at least 5 deletions for the derivative chromosomes of RNO1. These specified chromosome rearrangements appear to be non-random, as these aberrations are recurrently marked in the majority of analyzed karyotypes (Table 3). Furthermore, additional deletions and inversions in the karyotype cannot be excluded, since SKY analysis usually is not suitable to detect these specific types of aberrations.

It is noteworthy to mention that the recent analysis of the rat hepatic stellate cell line HSC-T6 that was established through SV40T transfection surprisingly showed relatively less chromosome changes without significant change in the modal chromosome number [6]. Originally, CFSC cells were spontaneously transformed from livers of carbon tetrachloride (CCl_4_)-induced cirrhotic male Wistar rats [7,8]. CCl_4_ is a non-genotoxic carcinogen known to be associated with intra-chromosomal recombination provoking liver cancer in rats and mice [29]. Therefore, the high frequency of chromosomal rearrangements might be caused by direct hepatotoxic effects of CCl_4_. The observation of multiple copies of chromosomes leading to a pseudo-triploid karyotype can be considered as a significant and specific feature of CFSC-2G cells, since similar multi-copy chromosomes are absent, for example, in the rat HSC-T6 cell line [6]. However, on the other hand, multiple copies of chromosomes were reported for immortalized mouse hepatocyte lines as well as for the human HSC line LX-2, which were all immortalized by SV40T [3,30]. Interestingly, we found no CFSC-2G cells without a Y chromosome or excess Y chromosomes. It may be worth speculating that the presence of one single Y chromosome has a protective effect for CFSC-2G and might be an advantage to maintain the immortalized phenotype. In humans, the loss of chromosome Y has been associated with aging and several diseases, such as cancer, demonstrating that this chromosome has functions beyond the male reproduction system [31,32,33]. Moreover, it is well-accepted that the Y chromosome in animals and humans contributes to essential cellular functions outside the testis, including transcription, translation, and protein stability [34].

Essentially, the karyotype abnormality described in this study provides insight to the genomic instability of CFSC-2G. The numerical and sweeping chromosomal changes might contribute to the capacity of this cell line to proliferate indefinitely.

Altogether, chromosome banding and multicolor SKY analysis revealed that CFSC-2G cells are characterized by extensive aneuploidy and striking chromosomal rearrangements that might be too complex to be used in daily routine for cell authentication. Therefore, we next performed STR DNA profile analysis involving the simultaneous amplification of 31 rat-specific STR markers. This analytical DNA technique is suitable to establish a characteristic fingerprint of these hypervariable microsatellite regions, allowing unambiguous cell authentication. The analysis revealed a unique STR profile in which 18 markers produced only one peak in the analysis and can be considered homozygous, while 13 markers gave rise to two peaks and can therefore be considered to be homozygous (Appendix A). The final STR profile confirmed that this cell line is of rat origin and revealed no indication for any mammalian interspecies contamination. Interestingly, only 2 of the 31 markers (i.e., markers 22 and 27) had the same allele constellation as those that we recently reported for HSC-T6 cells (Table 4) [6].

In sum, the presented biochemical characteristics, standard karyotyping analysis, SKY analysis, and the STR profile established in this study are certainly suitable means that can be used for cell authentication of CFSC-2G cells. In the future, this information will increase the biomedical value of CFSC-2G cells in liver research. In addition, our study helpful to fulfill the requirement of increasing the number of scientific journals and funding agencies that have started to require cell authentication prior to paper or grant submission when working with cell lines.

## 5. Conclusions

CFSC-2G cells are an attractive cellular model to analyze aspects of HSC biology. Since the original establishment of these cells three decades ago in the Marion Bessin Liver Research Center (Albert Einstein College of Medicine, Bronx, New York, NY, USA), the cells have made their way into many laboratories worldwide. The identification of specific chromosomal alterations and the definition of a unique STR profile will now allow discriminating these cells from all other cell lines. The demonstration that CFSC-2G cells express typical markers of HSCs (e.g., α-SMA, collagen type I, Vimentin, and Fibronectin), have a robust microfilament network, and can uptake and store fats confirms the proposed activated HSC phenotype. Altogether, the identification of a highly characteristic genetic profile will further increase the biomedical value of CFSC-2G for liver research. Together with cell line HSC-T6 that was recently genetically characterized by our team, researchers have now two independent, genetically well-characterized cell lines in hand that can be mutually used to reproduce experimental findings in continuous growing HSC lines of rat origin.

## Figures and Tables

**Figure 1 cells-11-02900-f001:**
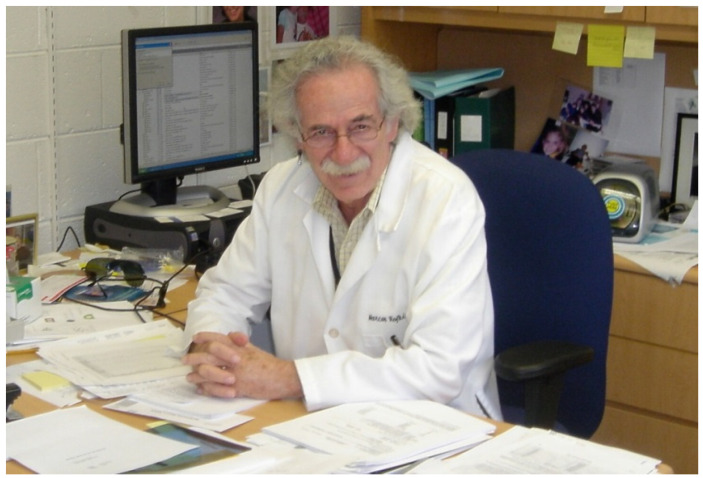
Marcus Rojkind, an outstanding expert in alcohol-induced liver disease research. Marcus Rojkind (29 July 1935–10 September 2011) was born in Mexico City and taught at the Marion Bessin Liver Research Center of the Albert Einstein College of Medicine (The Bronx, NY, USA) and researched in the Department of Biochemistry and Molecular Biology at the George Washington University Medical Center (Washington, DC, USA). Colleagues remember him as “a serious scientist, a careful experimenter, a critical colleagues, and an original thinker, respectful of his teachers and demanding (but generous) with his students” [14]. He readily provided biological materials for interested colleagues. Courtesy of Dr. Karina Reyes-Gordillo and Dr. Ruchi Shah.

**Figure 2 cells-11-02900-f002:**
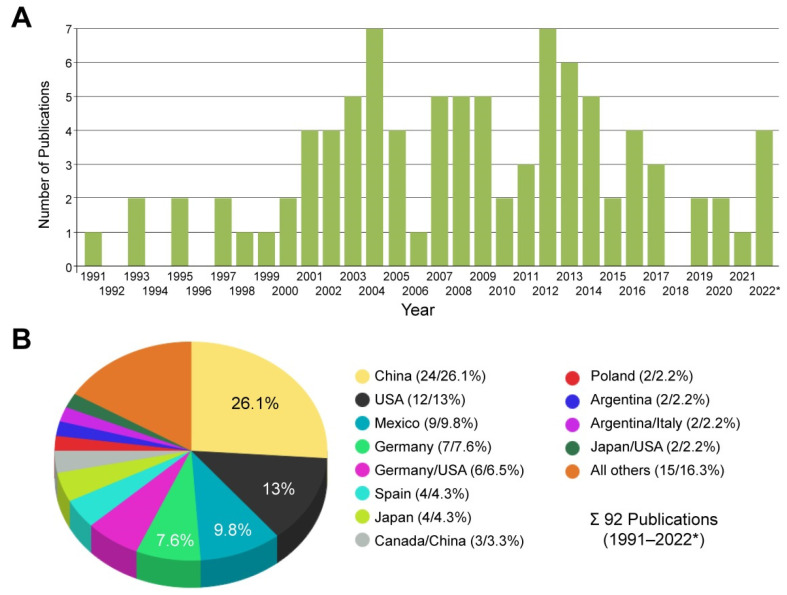
Studies using CFSC in articles published during 1991–2022. (**A**) Since the establishment of CFSC, reports using this cell line are published on a regular basis. Studies using CFSC were identified by searching the PubMed database or the Google search engine. The search was conducted on 19 August 2022. (**B**) Most studies that used CFSC were conducted in China (26.1%), USA (13%), Mexico (9.8%), and Germany (7.6%). For details see also Appendix A. The cell line was also used in joint studies from laboratories located in different countries. * Please note that papers for 2022 only cover the period from January to the middle of August.

**Figure 3 cells-11-02900-f003:**
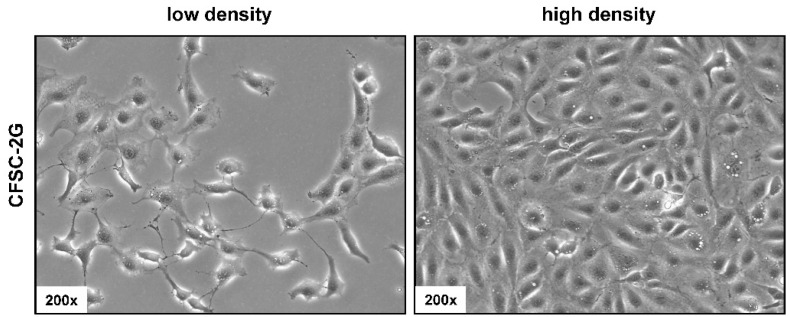
Light microscopic appearance of CFSC-2G cells. Cells were seeded in cell culture dishes and representative images taken at different cell densities at magnifications of 200×. Please note the extensive protrusions of the cells when cultured at low density.

**Figure 4 cells-11-02900-f004:**
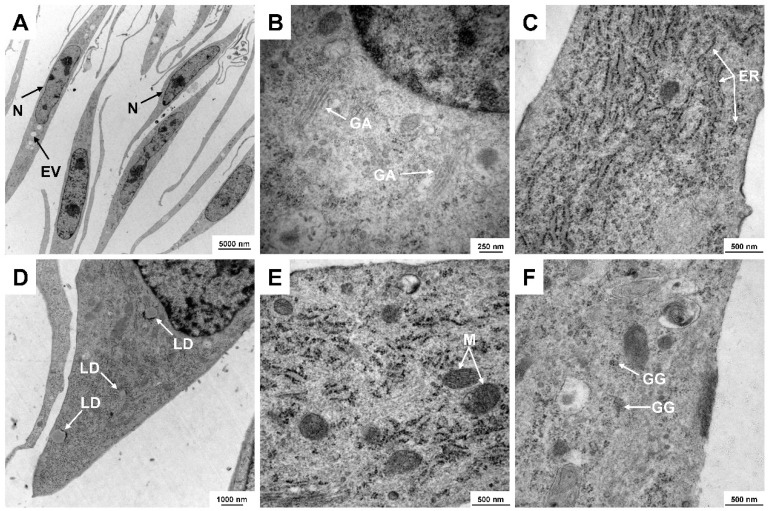
Representative electron microscopic analysis of CFSC-2G. Typical ultrastructural features of this cell line are (**A**) large nuclei (N) and endolysomal vesicles (EV), (**B**) prominent Golgi apparatus (GA), (**C**) remarkable rough endoplasmic reticulum (ER), (**D**) intracellular lipid droplets (LD), (**E**) large mitochondria (M), and (**F**) glycogen granules (GG). Original magnifications: (**A**) 2156×, (**B**) 35,970×, (**C**) 27,800×, (**D**) 7750×, and (**E**,**F**) 27,800×.

**Figure 5 cells-11-02900-f005:**
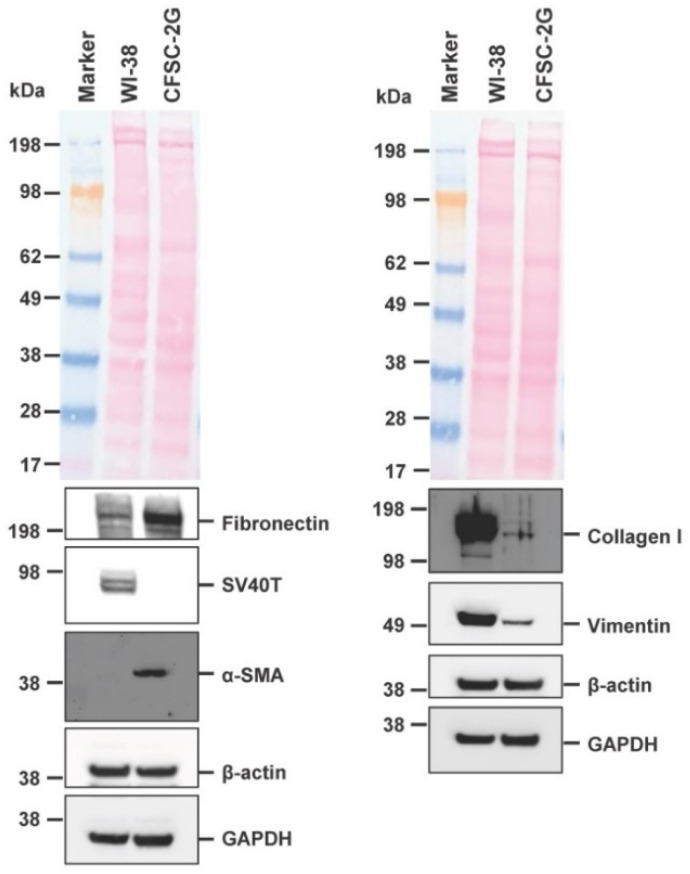
Expression of typical hepatic stellate cell markers in CFSC-2G. Protein extracts were prepared from WI-38 and CFSC-2G cells and analyzed for expression of Fibronectin, Simian virus 40 large T antigen (SV40T), α-smooth muscle actin (α-SMA), collagen type I, and Vimentin. The membranes were re-probed with antibodies directed against β-actin and GAPDH. In addition, the Ponceau S stains of membranes are shown to document equal protein loading in each lane.

**Figure 6 cells-11-02900-f006:**
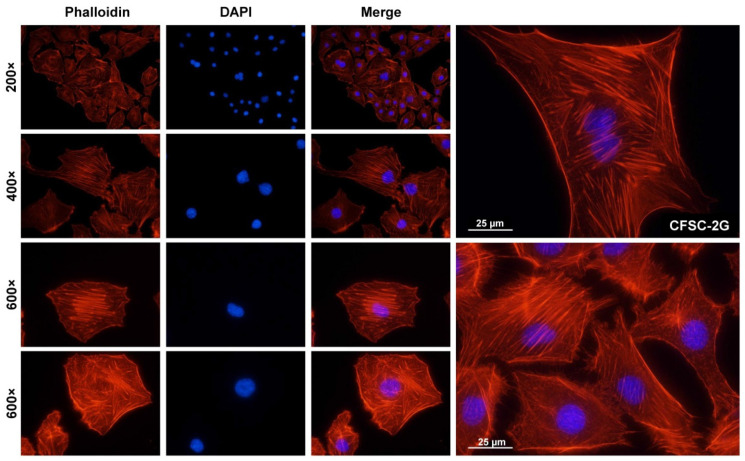
Cytoplasmic microfilaments in CFSC-2G. Cultured CFSC-2G cells were seeded on glass coverslips and stained with a Rhodamine–Phalloidin conjugate one day later. The prominent network of intracellular microfilaments points to a high activity in cytokinesis, movement, and cell motility. Nuclei were counterstained with 4′,6-diamidino-2-phenylindole (DAPI) and pictures were taken under a fluorescence microscope at indicated magnifications.

**Figure 7 cells-11-02900-f007:**
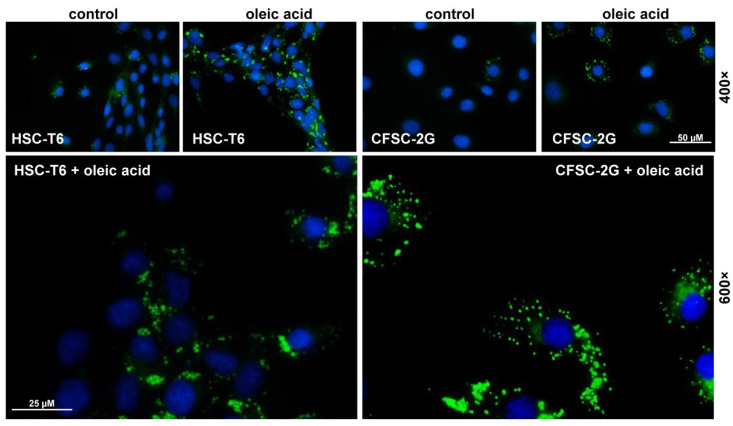
Lipid droplets in CFSC-2G. Cultured CFSC-2G cells were seeded on glass coverslips and grown in the presence of oleic acid and stained with BODIPY^TM^ 493/503. Cells that were stimulated with the vehicle (i.e., phosphate-buffered saline solution) served as controls. In addition, the same set of stimulation was performed in rat HSC line HSC-T6, which is an established immortalized HSC model for hepatic lipid and retinoid metabolism [10]. Nuclei were counterstained with 4′,6-diamidino-2-phenylindole (DAPI), glass coverslips mounted, and picture taken under a fluorescence microscope at magnification 400× or 600×.

**Figure 8 cells-11-02900-f008:**
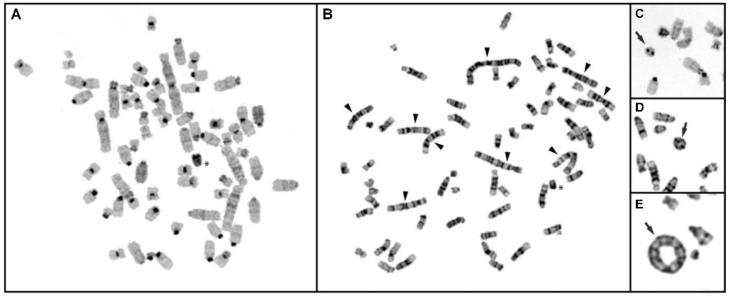
Conventional cytogenetic analysis of rat CFSC-2G cells. (**A**) CBG-banded metaphase spread displaying a complete heterochromatic Y chromosome (*). (**B**) GTG-banded metaphase showing numerous rearranged chromosomes (arrowheads). (**C**) CBG- as well as (**D**,**E**) GTG-banded partial metaphases showing ring chromosomes (arrows).

**Figure 9 cells-11-02900-f009:**
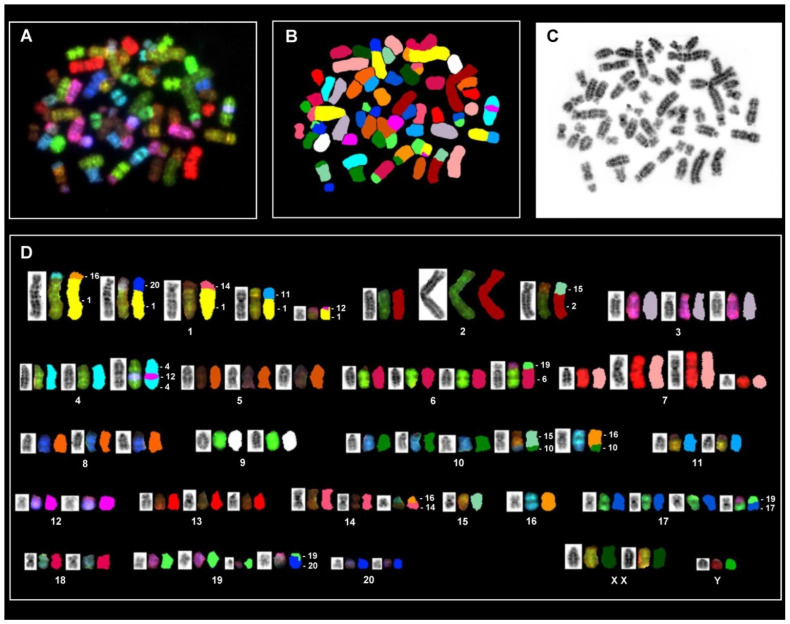
Spectral karyotyping of a rat CFSC-2G cell metaphase. (**A**) RGB image after hybridization with the SKY probe cocktail. (**B**) Classified pseudo-colored image of the same metaphase. (**C**) Inverted DAPI-stained image. (**D**) Karyotype of the metaphase showing spectrally classified, pseudo-colored chromosomes (**right**) compared with its inverted DAPI-stained chromosomes (**left**) and corresponding RGB image (**middle**). Specific chromosomes involved in the rearranged chromosomes are indicated. Based on the SKY analysis the CFSC-2G karyotype can be described as: <62>, XXY, +1, der(1)t(1;11)t(1;12)t(1;14)t(1;16)t(1;20), +2, der(2)i(2)(q)t(2;15), +3, +4, der(4)t(4;12), +5, +6, der(6)t(6;19), +7, der(7)i(7)(q)x2, +8, +10, der(10)t(10;15)t(10;16), +13, +14, der(14)t(14;16),−15,−16,+17, der(17)t(17;19), +19,der(19)t(19;20). Note (+) indicates the presence of multiple copies of specific chromosomes, while (−) indicates a deletion of a specific chromosome.

**Table 1 cells-11-02900-t001:** Antibodies used for Western blot analysis.

Antibody	Cat. No.	Company	Expected Size (kDa)	Dilution	Clonality
α-SMA	CBL171-I	Sigma-Aldrich, Taufkirchen, Germany	45	1000	r mAb
β-actin	A5441	Sigma-Aldrich	43	10,000	m mAb
Collagen I	NB600-408	Novus Biologicals, Wiesbaden, Germany	139	1000	r mAb
Fibronectin	AB1954	Sigma-Aldrich	262	3000	r pAb
GAPDH (6C5)	sc-32233	Santa Cruz Biotech., Santa Cruz, CA, USA	37	1000	m mAb
SV40T (v-300)	sc-20800	Santa Cruz	94	1000	r pAb
Vimentin	ab92547	Abcam, Berlin, Germany	54	3000	r mAb
g anti-rIgG (H + L), HRP	#31460	Invitrogen, ThermoFisher, Dreieich, Germany	NA	5000	g pAb
g anti-mIgG (H + L), HRP	#31430	Invitrogen	NA	5000	g pAb

Abbreviations used are: g, goat; m, mouse; r, rabbit; IgG, immunoglobulin G; mAb, monoclonal antibody; pAb, polyclonal antibody; NA, not applicable.

**Table 2 cells-11-02900-t002:** Gene expression in CFSC-2G determined by next-generation sequencing.

Transcript Id	Gene Id	Gene	Gene Description	TPM (CFSC-2G/HSC-T6)	Remarks *
ENSRNOT00000083468.1	ENSRNOG00000058039.1	Acta2	actin alpha 2, smooth muscle	10.9245/2.377	HSC marker
ENSRNOT00000005311.6	ENSRNOG00000003897.6	Col1a1	collagen type I alpha 1 chain	1408.42/2033.74	HSC marker
ENSRNOT00000004956.4	ENSRNOG00000003357.4	Col3a1	collagen type III alpha 1 chain	64.4932/2128.9	HSC marker
ENSRNOT00000024430.5	ENSRNOG00000018087.5	Vim	Vimentin	3853.69/2054.81	HSC marker
ENSRNOT00000057585.4	ENSRNOG00000014288.8	Fn1	Fibronectin 1	1383.81/143.953	HSC marker
ENSRNOT00000019772.6				702.835/352.585	
ENSRNOT00000013538.6	ENSRNOG00000009884.6	Lgals1	galectin 1	8557.92/4801.86	HSC marker
ENSRNOT00000017486.7	ENSRNOG00000012840.7	Sparc	secreted protein acidic and cysteine rich	1040.23/2188.34	HSC marker
ENSRNOT00000067011.2	ENSRNOG00000003772.7	Csrp2	cysteine and glycine-rich protein 2	62.7395/195.244	HSC marker
ENSRNOT00000013745.7	ENSRNOG00000010208.7	Timp1	TIMP metallopeptidase inhibitor	714.162/5518.57	inhibitor of MMPs
ENSRNOT00000004290.4	ENSRNOG00000003148.5	Timp2	TIMP metallopeptidase inhibitor 2	1030.84/369.667	inhibitor of MMPs
ENSRNOT00000010180.5	ENSRNOG00000007650.5	Cd63	Cd63 molecule	2732.07/1639.48	TIMP1 receptor
ENSRNOT00000011208.7	ENSRNOG00000008301.7	Tagln2	transgelin 2	642.605/754.93	smooth muscle marker
ENSRNOT00000015962.6	ENSRNOG00000020660.7	Cfl1	cofilin 1	1961.22/1670.37	cytoskeleton
ENSRNOT00000042459.4	ENSRNOG00000034254.4	Actb	actin, beta	3869.05/4379.5	cytoskeleton
ENSRNOT00000080216.1				1503.75/1820.7	
ENSRNOT00000078250.1	ENSRNOG00000056836.1	Cav1	caveolin 1	486.234/167.988	scaffolding protein
ENSRNOT00000008659.4	ENSRNOG00000009972.7	Rara	retinoic acid receptor, alpha	26.5339/24.8697	nuclear receptor
ENSRNOT00000084644.1				20.4423/23.8544	
ENSRNOT00000033048.6	ENSRNOG00000024061.7	Rarb	retinoic acid receptor, beta	0.00872314/0.541197	nuclear receptor
ENSRNOT00000016801.5	ENSRNOG00000012499.7	Rarg	retinoic acid receptor, gamma	49.0507/64.5919	nuclear receptor
ENSRNOT00000017096.7				25.6328/39.1824	
ENSRNOT00000012892.4	ENSRNOG00000009446.4	Rxra	retinoid X receptor alpha	41.2102/18.9052	nuclear receptor
ENSRNOT00000041613.5	ENSRNOG00000000464.7	Rxrb	retinoid X receptor beta	15.9738/20.0384	nuclear receptor
ENSRNOT00000087670.1				1.53037/1.79029	
ENSRNOT00000086978.1				1.02833/1.08069	
ENSRNOT00000081588.1				1.61434/0.804912	
ENSRNOT00000091182.1				0.347194/0.723898	
ENSRNOT00000087895.1				0.0953106/0.688452	
ENSRNOT00000084638.1				0.615142/0.499241	
ENSRNOT00000012137.5 ENSRNOT00000051858.5 ENSRNOT00000082969.1	ENSRNOG00000008839.7	Pparg	peroxisome proliferation-activated receptor-γ	0.883663/0.157304 0.386135/0.248569 0/6.50011	nuclear receptor
ENSRNOT00000077227.1	ENSRNOG00000004537.6	Rxrg	retinoid X receptor gamma	0/0	nuclear receptor
ENSRNOT00000006117.5				0.271271/0	
ENSRNOT00000019571.3	ENSRNOG00000014090.3	Retsat	retinol saturase	58.3177/52.476	retinoid metabolism
ENSRNOT00000081756.1	ENSRNOG00000053850.1	Rdh5	retinol dehydrogenase 5	0.693011/0.0383589	retinoid metabolism
ENSRNOT00000032076.4	ENSRNOG00000025767.4	Rdh8	retinol dehydrogenase 8	0.0525309/0.112568	retinoid metabolism
ENSRNOT00000009096.4	ENSRNOG00000006681.4	Rdh10	retinol dehydrogenase 10	21.0453/3.87175	retinoid metabolism
ENSRNOT00000085927.1	ENSRNOG00000054770.1	Rdh11	retinol dehydrogenase 11	35.9222/26.8725	retinoid metabolism
ENSRNOT00000078436.1				0.245147/0.15904	
ENSRNOT00000089162.1	ENSRNOG00000056553.1	Rdh12	retinol dehydrogenase 12	0/1.06243	retinoid metabolism
ENSRNOT00000031462.5	ENSRNOG00000027919.5	Rdh13	retinol dehydrogenase 13	1.19281/1.96789	retinoid metabolism
ENSRNOT00000006020.6	ENSRNOG00000039551.3	Rdh14	retinol dehydrogenase 14	19.6884/8.16116	retinoid metabolism
ENSRNOT00000093003.1				5.89539/2.97058	
ENSRNOT00000092948.1				1.27751/0.857514	
ENSRNOT00000018622.4	ENSRNOG00000013794.4	Rbp1	retinol-binding protein 1	0/25.3842	retinoid metabolism
ENSRNOT00000018755.6	ENSRNOG00000013932.6	Rbp2	retinol-binding protein 2	1.64899/0.309974	retinoid metabolism
ENSRNOT00000021055.7	ENSRNOG00000015518.7	Rbp4	retinol-binding protein 4	0.0677519/0.0363529	retinoid metabolism
ENSRNOT00000021348.5	ENSRNOG00000015850.5	Rbp7	retinol-binding protein 7	0.398306/0.244971	retinoid metabolism
ENSRNOT00000082156.1	ENSRNOG00000025608.4	Lrat	lecithin retinol acyltransferase	0.0883818/0.0125274	retinoid metabolism
ENSRNOT00000022113.4	ENSRNOG00000016275.4	Ttr	transthyretin	0.049935/0.161195	RBP transporter
ENSRNOT00000030919.5	ENSRNOG00000022619.6	Fth1	ferritin heavy chain 1	5047.69/9890.55	iron storage
ENSRNOT00000087162.1	ENSRNOG00000059900.1	Bst2	bone marrow stromal cell antigen 2	291.2/1681.66	antiviral response
ENSRNOT00000091906.1				127.083/589.283	
ENSRNOT00000080988.1	ENSRNOG00000052802.1	Aldoa	Aldolase, fructose-bisphosphate	1914.86/1471.21	glycolysis
ENSRNOT00000087928.1				745.559/492.926	
ENSRNOT00000088473.1				308.32/183.94	
ENSRNOT00000015332.7	ENSRNOG00000011329.7	Pkm	pyruvate kinase M1/2	1952.24/1437.52	glycolysis
ENSRNOT00000083666.1				596.921/455.466	
ENSRNOT00000077604.1	ENSRNOG00000058249.1	Pgk1	Phosphoglycerate kinase 1	2609.96/1404.31	glycolysis
ENSRNOT00000085653.1				0/0.204793	
ENSRNOT00000050443.4	ENSRNOG00000018630.7	LOC108351137	Glyceraldehyde-3-phosphate dehydrogenase	3223.93/4598.86	glycolysis
ENSRNOT00000041328.3	ENSRNOG00000030963.3			2962.7/4243.2	

* Abbreviations used: HSC, hepatic stellate cell; MMP, matrix metalloproteinases; RBP, retinol-binding protein.

**Table 3 cells-11-02900-t003:** Common and sporadic structural chromosomal aberrations (●) in CFSC-2G identified through spectral karyotyping.

Cell	Chromosome Numbers	der(1)t(1;11)	der(1)t(1;12)	der(1)t(1;14)	der(1)t(1;16)	der(1)t(1;20)	rob(2;2)	der(2)t(2;5)	der(2)t(2;6)	der(2)t(2;12)	der(2)t(2;15)	der(2)t(2;17)	der(3)t(3;7)	der(4)t(4;12)	der(4)t(4;12;16	der(5)t(5;Y)	der(6)t(6;19)	der(6)t(7;6;19)	der(7)t(7;10)	der(8)t(8;15)	der(9)t(9;12)	der(10)t(10;Y)	der(10)t(10;15)	der(10)t(10;16)	der(12)t(12;19)	der(12(t(12;20)	der(13)t(13;18)	der(14)t(14;16)	der(15)t(15;19)	der(17)t(17:19)	der(18)t(18;20)	der(19)t(19;20)
1	63, XXY	●	●	●	●	●	●				●			●				●				●	●	●		●		●		●		●
2	62, XXY	●	●	●		●						●		●		●							●	●	●	●		●		●		●
3	62, XXY	●	●	●	●	●	●				●			●			●						●	●				●		●		●
4	62, XXY	●	●	●	●	●					●			●					●		●		●	●		●	●	●		●		●
5	58, XXY	●	●	●	●	●	●				●			●										●				●			●	
6	62, XXYY	●	●	●		●				●	●		●	●									●	●				●		●		●
7	57, XX	●	●		●	●	●							●			●							●				●		●		●
8	65, XXY	●	●	●	●	●	●				●			●	●		●						●	●				●		●		●
9	63, XXY	●	●	●	●	●	●				●			●			●			●				●				●		●		●
10	64, XXY	●	●	●	●	●	●		●					●					●					●				●	●	●		●
11	64, XXY	●	●	●	●	●	●	●			●			●									●	●				●		●		●

**Table 4 cells-11-02900-t004:** STR profile of CFSC-2G using the 31 species-specific STR markers.

SN	Marker Name	Chromosomal Location	Allele Sizes (bp) in CFSC-2G	Allele Sizes (bp) in HSC-T6 *
1	73	1	194, 203	194
2	8	2	236	234
3	2	2	126	127
4	4	3	268, 270	238
5	3	3	160, 182	160, 162
6	26	4	150	166
7	19	4	180	175
8	81	5	130, 134	130, 132
9	34	6	184, 189	188
10	30	7	188, 192	186, 192
11	24	8	260	249, 253
12	59	9	145	143, 146, 180
13	62	9	166	177
14	1	10	105	96
15	55	10	210, 214	210, 218
16	36	11	222	234
17	67	11	154, 156	165
18	13	12	121	121, 135
19	35	13	197	197, 203
20	42	13	125	127
21	70	14	158, 175	175, 179
22	61	15	128	128
23	79	15	172, 180	172
24	90	16	159, 161	174
25	69	16	138	139
26	78	17	136, 151	147, 151
27	15	18	232	232
28	16	18	251, 260	247, 251
29	75	19	144	144, 184
30	96	20	210	210, 212
31	91	20	221	205, 211

Profiling was performed using the CellCheck^TM^ Rat Panel (IDEXX BioAnalytics, Columbia, MO, USA). * STR data were taken from [6]. Abbreviations used: bp, base pairs; SN, serial number; STR, short tandem repeat. The individual electropherograms of each STR marker are depicted in Appendix A.

## Data Availability

This manuscript contains most of the original data generated during our study. However, additional datasets (e.g., the complete fastq data file of our NGS analysis, additional illustrations of the SKY painting) can be requested from the corresponding author.

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
