# Peer review of "Rat Hepatic Stellate Cell Line CFSC-2G: Genetic Markers and Short Tandem Repeat Profile Useful for Cell Line Authentication"

_cells, 2022, doi:10.3390/cells11182900_

Round 1
Reviewer 1 Report
This is a useful paper to provide information on the identification of non-immortalized rat HSC line CFSC. The authors are familiar with cell authentication. There are a few comments that this reviewer hopes the authors to address.
Comments:
1. Figure 7. In lipid droplets, is retinoid also stored? After treating lipids, do HSCs become a more quiescent phenotype? For example, is PPARg expression increased?
2. The subtitle of “3.6. Genetic analysis of HSC-T6 Cells” Is this typo?
3. What is the expression level of PPARg, which is a marker for quiescent HSCs?
Author Response
Dear Reviewer 1,
many thanks for reviewing our paper. Please find our comments to your suggestions in the attached pdf-file.
Many thanks
Ralf Weiskirchen
(on behalf of all authors)

Reviewer 2 Report
This manuscript by Nanda et al. renders an insightful notion to unravel the nature of the CFSC-2G cells from cirrhotic rat livers induced by carbon tetrachloride. They described comprehensive molecular features of CFSC-2G cells similar to T6 cells and no SV40 large T antigen expression. They also showed CFSC-2G genome has accumulated extensive chromosome rearrangements and most chromosomes exist in multiple copies producing a pseudo-triploid karyotype as well as 31 species-specific short tandem repeats by NGS. I have to appreciate the authors giving us a picture in memory of Prof. Marcus Rojkind in figure 1, which reminds the younger scientist of the importance of being a serious scientist, a careful experimenter, a critical colleague, and an original thinker, respectful of his teachers and demanding (but generous) with his students. I just have only a few commands.
1. Could the authors give us a list that summarizes (such as a table) the advantages and disadvantages of choosing the common HSC cell lines for the different models of liver experiments? Which one is better than the other in vitro study or we should choose at least two cells line for confirmation? Is CFSC-2G superior to others due to it does not have an SV40 large T antigen?
2. Why STR is important for cell authentication, or it affects the activation of HSC? What does it mean “the result of this analysis shows that CFSC-2G had a unique STR profile that was markedly different from that we have recently established for rat HSC line HSC-T6?” Does it affect the nature of HSC?
Minor
1. Line 44, change “ the can” to “ they can”
Author Response
Dear Reviewer 2,
many thanks for reviewing our paper. Please find attached our comments to your suggestions in the attached pdf-file.
Regards
Ralf Weiskirchen
(on behalf of all authors)
